# Efficient nonmyopic batch active search

**Shali Jiang**
CSE, WUSTL
St. Louis, MO 63130
jiang.s@wustl.edu

**Gustavo Malkomes**
CSE, WUSTL
St. Louis, MO 63130
luizgustavo@wustl.edu

**Matthew Abbott**
CSE, WUSTL
St. Louis, MO 63130
mbabbott@wustl.edu

**Benjamin Moseley**
Tepper School of Business, CMU and
Relational AI
Pittsburgh, PA 15213
moseleyb@andrew.cmu.edu

**Roman Garnett**
CSE, WUSTL
St. Louis, MO 63130
garnett@wustl.edu

## Abstract

Active search is a learning paradigm for actively identifying as many members of a given class as possible. A critical target scenario is high-throughput screening for scientific discovery, such as drug or materials discovery. In these settings, specialized instruments can often evaluate *multiple* points simultaneously; however, all existing work on active search focuses on sequential acquisition. We bridge this gap, addressing batch active search from both the theoretical and practical perspective. We first derive the Bayesian optimal policy for this problem, then prove a lower bound on the performance gap between sequential and batch optimal policies: the "cost of parallelization." We also propose novel, efficient batch policies inspired by state-of-the-art sequential policies, and develop an aggressive pruning technique that can dramatically speed up computation. We conduct thorough experiments on data from three application domains: a citation network, material science, and drug discovery, testing all proposed policies with a wide range of batch sizes. Our results demonstrate that the empirical performance gap matches our theoretical bound, that nonmyopic policies usually significantly outperform myopic alternatives, and that diversity is an important consideration for batch policy design.

## 1   Introduction

In active search (AS), we seek to sequentially inspect data to discover as many members of a desired class as possible with a limited budget. Formally, suppose we are given a finite domain of $n$ elements $\mathcal{X} = \{x_i\}_{i=1}^n$, among which there is a rare, valuable subset $\mathcal{R} \subset \mathcal{X}$. We call the members of this class *targets* or *positive items.* The identities of the targets are unknown *a priori,* but can be determined by querying an expensive oracle that can compute $y = \mathbb{1}\{x \in \mathcal{R}\}$ for any $x \in \mathcal{X}$. Given a budget $T$ on the number of queries we can provide the oracle, we wish to design a policy that sequentially queries items $\{x_t\} = \{x_1, x_2, \ldots, x_T\}$ to maximize the number of targets identified, $\sum y_t$. Many real-world problems can be naturally posed in terms of active search; drug discovery [7, 17, 18], fraud detection, and product recommendation [21] are a few examples.

Previous work [6] has developed Bayesian optimal policies for active search with a natural utility function. Not surprisingly, this policy is computationally intractable, requiring cost that grows exponentially with the horizon. Therefore the optimal policy must be approximated in practice. Several approximation schemes for active search have been proposed and studied, including simple myopic lookahead approximations [6]. Jiang et al. [12] recently proposed an efficient, *nonmyopic* search (ENS) policy, and demonstrated this policy yields remarkable empirical performance on search

problems from various domains, including drug and materials discovery. Although these policies are empirically effective, there is a large theoretical gap between the performance of the optimal policy and any efficient approximation: Jiang et al. [12] have shown that it is impossible to $\alpha$-approximate the expected performance of the optimal policy for any constant $\alpha$ in polynomial time.

These previous investigations mentioned above all focused on *sequential active search* (SAS), where we query one point at a time. However, in many real applications, we can query a *batch* of multiple points simultaneously. For example, modern high-throughput screening technologies for drug discovery can process microwell plates containing 96+ compounds at a time. No policies designed for this batch active search setting are currently available. Previous work has produced batch policies for different active learning or search settings, which we will discuss in Section 4.

We investigate *batch active search* (BAS) from both the theoretical and practical perspectives. We first derive the Bayesian optimal policy for BAS, and show that its time complexity in general is dauntingly high, except in the trivial one-step (myopic) case. We then prove an asymptotic lower bound on the expected performance gap between the optimal sequential and batch policies.

Next we consider practical concerns such as effective policy design. We generalize the recently proposed efficient nonmyopic sequential policy (ENS) from [12] to the batch setting. The nonmyopia of ENS is automatically inherited, but efficiency is lost as the batch version involves combinatorial optimization (i.e., set function maximization). We propose and study two efficient approximation strategies. The first strategy is a sequential simulation, where we simulate sequential ENS to construct a batch using a fictional labeling oracle. The second strategy is greedily maximizing the marginal gain to our batch ENS score, motivated by our conjecture that the inherent batch score is submodular. We prove that sequential simulation of the one-step Bayesian optimal policy with a *pessimistic* oracle (i.e., one that always outputs negative labels) near-optimally maximizes the probability that *at least one point in the batch* is positive. This theoretical support of pessimism is in contrast to other settings such as Bayesian optimization, where pessimism has been used as a *heuristic* for batch policies.

We also improve the pruning techniques suggested by Jiang et al. [12] considerably to reduce the computational overhead of our proposed policies in practice. We demonstrate a connection with lazy evaluation [5] and show that our pruning strategy can provide a speedup of over 50 times in a drug discovery setting.

Finally, we conduct thorough experiments on data from three domains: a citation network, material science, and drug discovery. In total we study 14 policies: the one-step optimal batch policy, 12 sequential simulation policies (three sequential policies combined with four fictional oracles), and greedy maximization of the batch version of ENS. We observe that ENS-based (nonmyopic) policies almost always provide a significant improvement in performance. Two policies are particularly notable: sequential simulation of ENS with a pessimistic oracle and greedy maximization of batch ENS. The latter is shown to be more robust for larger batch sizes.

## 2 Bayesian optimal batch active search

We begin our investigation by establishing the *optimal* policy for batch active search using the framework of Bayesian decision theory. To cast batch active search into this framework, we express our preference over different datasets $\mathcal{D} = \{(x_i, y_i)\}$ through a natural utility: $u(\mathcal{D}) = \sum y_i$, which simply counts the number of targets in $\mathcal{D}$. Occasionally we will use the notation $u(Y)$ for $u(\mathcal{D})$ when $\mathcal{D} = (X, Y)$. We now consider the problem of sequentially choosing a set of $T$ (a given budget) points $\mathcal{D}$ with the goal of maximizing $u(\mathcal{D})$. In the batch setting, for each query we must select a batch of $b$ points and will then observe all their labels at the same time. We use $X_i = \{x_{i,1}, x_{i,2}, \ldots x_{i,b}\}$ to denote a batch of points chosen during the $i$th iteration, and $Y_i = \{y_{i,1}, y_{i,2}, \ldots y_{i,b}\}$ the corresponding labels. We use $\mathcal{D}_i = \{(X_k, Y_k)\}_{k=1}^{i}$ to denote the observed data after $i \le t$ batch queries, where $t = \lceil T/b \rceil$.

We assume a probability model $\mathcal{P}$ is given, providing the posterior marginal probability $\Pr(y \mid x, \mathcal{D})$ for any point $x \in \mathcal{X}$ and observed dataset $\mathcal{D}$. At iteration $i + 1$ (given observations $\mathcal{D}_i$), the Bayesian optimal policy chooses a batch $X_{i+1}$ maximizing the expected utility at termination, recursively assuming optimal continued behavior:

$$X_{i+1} = \arg\max_{X} \mathbb{E}\big[u(\mathcal{D}_t \setminus \mathcal{D}_i) \mid X, \mathcal{D}_i\big]. \tag{1}$$

Note that the additive nature of our chosen utility allows us to ignore the utility of the already gathered data in the expectation.

To derive the expected utility, we adopt the standard technique of backward induction, as used by for example Garnett et al. [6] to analyze the sequential case. The base case is when only one batch is left $(i = t - 1)$. The expected utility resulting from a proposed final batch $X$ is then

$$\mathbb{E}\big[u(\mathcal{D}_t \setminus \mathcal{D}_{t-1}) \mid X, \mathcal{D}_{t-1}\big] = \mathbb{E}_{Y \mid X, \mathcal{D}_{t-1}}\big[u(Y)\big] = \sum_{x \in X} \Pr(y = 1 \mid x, \mathcal{D}_{t-1}), \qquad (2)$$

where $\mathbb{E}_{Y \mid X, \mathcal{D}_i}$ is the expectation over the joint posterior distribution of $Y$ (the labels of $X$) conditioned on $\mathcal{D}_i$. In this case, designing the optimal batch (1) by maximizing the expected utility is trivial: we select the points with the highest probabilities of being targets, reflecting pure exploitation. This optimal batch can then be found in $\mathcal{O}(n \log b)$ time using, e.g., min-heap of size $b$.

In general, when $i \leq t - 1$, the expected terminal utility resulting from choosing a batch $X$ at iteration $i + 1$ and acting optimally thereafter can be written as a Bellman equation as follows:

$$\mathbb{E}\big[u(\mathcal{D}_t \setminus \mathcal{D}_i) \mid X, \mathcal{D}_i\big] = \sum_{x \in X} \Pr(y = 1 \mid x, \mathcal{D}_i) + \mathbb{E}_{Y \mid X, \mathcal{D}_i}\Big[\max_{X'} \mathbb{E}\big[u(\mathcal{D}_t \setminus \mathcal{D}_{i+1}) \mid X', \mathcal{D}_{i+1}\big]\Big],$$
$$(3)$$

where the first term represents the expected utility resulting immediately from the points in $X$, and the second part is the expected future utility from the following iterations.

The most interesting aspect of the Bayesian optimal policy is that these immediate and future reward components in (3) can be interpreted as automatically balancing exploitation (immediate utility) and exploration (expected future utility given the information revealed by the present batch).

However, without further assumptions on the joint label distribution $\mathcal{P}$, exact maximization of (3) requires enumerating the whole search tree of the form $\mathcal{D}_i \rightarrow X_{i+1} \rightarrow Y_{i+1} \rightarrow \cdots \rightarrow X_t \rightarrow Y_t$. The branching factor of the $X$ layers is $\binom{n}{b}$, as we must enumerate all possible batches. The branching factor of the $Y$ layers is $2^b$, as we must enumerate all possible labelings of a given batch. So the total complexity of a naïve implementation computing the optimal policy at iteration $i + 1$ would be a daunting $\mathcal{O}\big((2n)^{b(t-i)}\big)$. The running time analysis in [6] is a special case of this result where $b = 1$.

The optimal policy is clearly computationally infeasible, so we must resort to suboptimal policies to proceed in practice. One reasonable and practical alternative is to adopt a myopic lookahead approximation to the optimal policy. A *greedy* (one-step lookahead) approximation, which always maximizes the expected marginal gain in (2), constructs each batch by selecting the points with highest probability of being a target. We will refer to this policy as greedy-batch, and this will serve as a natural baseline batch policy for active search.

## 2.1 Adaptivity gap

For purely sequential policies (i.e., $b = 1$), every point is chosen based on a model informed by all previous observations. However, for batch policies ($b > 1$), points are typically chosen with less information available. For example, in the extreme case when $b = T$, every point in our budget must be chosen before we have observed anything, hence we might reasonably expect our search performance to suffer. Clearly there must be an inherent cost to batch policies compared to sequential policies due to a loss of adaptivity. How much is this cost?

We have proven the following lower bound on the inherent "cost of parallelism" in active search:

**Theorem 1.** *There exist active search instances with budget $T$, such that $\frac{\text{OPT}_1}{\text{OPT}_b}$ is $\Omega\big(\frac{b}{\log T}\big)$, where* $\text{OPT}_x$ *is the expected number of targets found by the optimal batch policy with batch size $x \geq 1$.*

*Proof sketch.* We construct a special type of active search instance where the location of a large trove of positives is encoded by a binary tree, and a search policy must take the correct path through the tree to decode a treasure map pointing to these points. We design the construction such that a sequential policy can easily identify the correct path by walking down the tree directed by the labels of queried nodes. A batch policy must waste a lot queries decoding the map as the correct direction is only revealed after constructing an entire batch. We show that even the optimal batch policy has a very low probability of identifying the location of the hidden targets quickly enough, so that the expected utility is much less than that of the optimal sequential policy. A detailed proof is given in the supplementary material. $\qquad \square$

Thus the expected performance ratio between optimal sequential and batch policies, also known as *adaptivity gap* in the literature [1], is lower bounded linearly in batch size. This theorem is not only of theoretical interest: it can also provide practical guidance on choosing batch sizes. Indeed, in drug discovery, modern high-throughput screening technologies provide many choices for batch sizes; understanding the inherent loss from choosing larger batch sizes provides valuable information regarding the tradeoff between efficiency and cost.

## 3  Efficient nonmyopic approximations

The greedy-batch policy is myopic in the sense that each decision represents pure exploitation: the future reward is always assumed to be zero, and the remaining budget is not taken into consideration. Here we will generalize a recently proposed nonmyopic sequential active search policy, ENS [12], to the batch setting and propose two techniques to approximately compute it.

Our proposed adaptation of ENS to batch setting can be motivated with the following question: how many targets would we expect to find if, after selecting the current batch, we spent the entire remaining budget simultaneously? If this were the case, then the maximum future utility could be computed without recursion:

$$\mathbb{E}[u(\mathcal{D}_t \setminus \mathcal{D}_i) \mid X, \mathcal{D}_i] = \sum_{x \in X} \Pr(y = 1 \mid x, \mathcal{D}_i) +$$
$$\mathbb{E}_{Y \mid X, \mathcal{D}_i}\big[\max_{X' : |X'| = T - b - |\mathcal{D}_i|} \mathbb{E}\big[u(Y') \mid X', \mathcal{D}_i, X, Y\big]\big]. \quad (4)$$

Note the optimal final action simply selects the points with the highest $T - b - |\mathcal{D}_i|$ probabilities, allowing the expected future reward to be computed exactly and efficiently. We may use this insight to rewrite (4) as (using $f(X \mid \mathcal{D}_i)$ as shorthand for the expected utility from selecting $X$ given $\mathcal{D}_i$):

$$f(X \mid \mathcal{D}_i) = \sum_{x \in X} \Pr(y = 1 \mid x, \mathcal{D}_i) + \mathbb{E}_{Y \mid X, \mathcal{D}_i}\left[\sum_{T - b - |\mathcal{D}_i|}' \Pr\left(y' = 1 \mid x', \mathcal{D}_i, X, Y\right)\right]. \quad (5)$$

Here we have adopted the notation $\sum_s'$ from [12] to denote the sum of the top $s$ probabilities over the unlabeled points, $x' \in \mathcal{X} \setminus (\mathcal{D}_i \cup X)$. Jiang et al. [12] gave a further interpretation of the ENS policy as approximating the optimal expected utility (3) by assuming that the remaining unlabeled points after this batch are conditionally independent, so that there is no need to recursively enumerate the search tree. This assumption might seem unrealistic at first, but when many *well-spaced* points are observed, we note they might approximately "D-separate" the remaining unlabeled points. Further, ENS naturally encourages the selection of well-spaced points (targeted exploration) in the initial state of the search [12].

The nonmyopia of (5) is automatically inherited in generalizing from sequential to batch setting due to explicit budget awareness. Unfortunately, the efficiency of the sequential ENS policy is not preserved. Direct maximization of (5) still requires combinatorial search over all subsets of size $b$. Moreover, to evaluate a given batch, we need to enumerate all its possible labelings ($2^b$ in total) to compute the expectation in the second term. Accounting for the cost of conditioning and summing the top probabilities, the total complexity would be $\mathcal{O}\big((2n)^b \, n \log T\big)$.

We propose two strategies to tackle these computational problems below.

### 3.1  Sequential simulation

The cost of computing the proposed batch policy has exponential dependence on the batch size $b > 1$. To avoid this, our first idea is to reduce BAS to SAS ($b = 1$). We select points one at a time to add to a batch by maximizing the sequential ENS score (i.e., (5) with $b = 1$). We then use some fictional labeling oracle $\mathcal{L} \colon \mathcal{X} \to \{0, 1\}$ to simulate its label and incorporate the observation into our dataset. We repeat this procedure until we have selected $b$ points. Note that we could use this basic construction replacing ENS by any other sequential policy $\pi$, such as the one-step or two-step Bayesian optimal policies [6].

We will see that the behavior of the fictional labeling oracle has large influence on the behavior of resulting search policies. Here we will consider four fictional oracles: (1) sampling, where we randomly sample a label from its marginal distribution; (2) most-likely, where we assume the most-likely label; (3) pessimistic, where we always believe all labels are negative; and (4) optimistic, where always believe all labels are positive.

Sequential simulation is a common *heuristic* in similar settings like batch Bayesian optimization, as we will discuss in detail in the next section. Here we provide some mathematical rationale of this procedure in a special case: the one-step optimal (greedy) search policy combined with the pessimistic oracle. This proposition is inspired by the work of Wang [23].

**Proposition 1.** *The batch constructed by sequentially simulating the greedy active search policy with a pessimistic oracle near-optimally maximizes the probability that* at least one *of the points in the batch is positive, assuming that marginal target probabilities of unlabeled points are nonincreasing when conditioning on a negative observation.*

*Proof sketch.* We show that the probability of a batch having at least one positive is a monotone submodular set function, and that sequentially simulating the one-step policy with the pessimistic oracle equivalently maximizes the marginal gain of this function. Therefore, it is near-optimal [16]. See the supplementary materials for a formal proof.

Note the assumption in this proposition simply means the probability model does not involve negative label correlations; the $k$-nn model used in our experiments satisfies this assumption.

With this result, it is not hard to see that sequentially simulating the greedy policy with an optimistic oracle greedily maximizes the probability that *all* points in the batch are positive. In this case, however, the corresponding set function is not submodular so we don't know if there are optimality guarantees.

Note we are not claiming that the objective of finding at least one positive serves as a good basis for batch active search; actually as we will see in our experiments, this is often much worse than other nonmyopic batch policies we propose. However, we believe this result provides theoretical insight that could shed light on other batch policies under similar settings. For example, this policy can be considered as an active search counterpart of a batch version of *probability of improvement* for Bayesian optimization [13].

### 3.2 Greedy approximation

Our second strategy is motivated by our conjecture that (5) is a monotone submodular function under reasonable assumptions. If that is the case, then again a *greedy* batch construction returns a batch with near-optimal score [16]. We therefore propose to use a greedy algorithm to sequentially construct the batch by maximizing the marginal gain. That is, we begin with an empty batch $X = \emptyset$. We then sequentially add $b$ points by adding the point maximizing the marginal gain:

$$x = \arg\max_x \Delta_f(x \mid X), \tag{6}$$

where

$$\Delta_f(x \mid X) = f(X \cup \{x\} \mid \mathcal{D}_i) - f(X \mid \mathcal{D}_i). \tag{7}$$

When $b$ is large, this procedure is still expensive to compute due to the expectation term in (5), requiring $\mathcal{O}(2^b)$ operations to compute exactly. Here we approximate the expectation using Monte Carlo sampling with a small set of samples of the labels. Specifically, given a batch of points $X$, we approximate (5) with samples $S = \{\tilde{Y} : \tilde{Y} \sim Y \mid X, \mathcal{D}_i\}$:

$$f(X \mid \mathcal{D}_i) \approx \sum_{x \in X} \Pr(y = 1 \mid x, \mathcal{D}_i) + \frac{1}{|S|} \sum_{Y \in S} \left[ \sum'_{T-b-|\mathcal{D}_i|} \Pr(y' = 1 \mid x', \mathcal{D}_i, X, Y) \right]. \tag{8}$$

We will call the batch policy described above batch-ENS. Note batch-ENS using *one* sample of the labels in a batch is similar to sequential simulation of ENS with the sampling oracle, though the two policies are motivated in different ways.

### 3.3 Implementation and pruning

We adopt $k$-nn as our probability model, so the tricks described in 3.2 of [12] can be adopted. The time complexity per iteration for sequential simulation of ENS is $\mathcal{O}(n(\log n + m \log m + T))$, where $n$ is the total number of points, $m$ is the maximum number of points that can be influenced by any point with the $k$-nn model, and $T$ is the total budget. For batch-ENS, the time complexity at each iteration is also the same as ENS for each sample, so the complexity increases linearly as the number of samples. Note with $2^s$ samples, (8) can be exactly computed for the first $s + 1$ points; so the complexity for selecting the $j$th point of a batch would be $\mathcal{O}\big(\min(2^{j-1}, 2^s)\, n(\log n + m \log m + T)\big)$.

We also improve the bounding and pruning strategy developed in [12], and our new procedure is now similar in spirit to *lazy evaluation* [5]. On drug discovery datasets, on average over 98% of the candidate points can be pruned in each iteration, a speedup of over 50 times. Details and results regarding the effectiveness of pruning in practice can be found in the supplemental material.

# 4 Related work

Active search (AS) and its variants have been the subject of a great deal of recent work [6, 7, 12, 14, 15, 22, 24–27]; nevertheless, to the best of our knowledge, this is the first study on batch active search under this particular setting.

Warmuth et al. [25, 26] considered a different goal of batch active search: to find all or a given number of actives as soon as possible. Our goal, in contrast, is to find as many actives as possible in a given budget, which encourages more nonmyopic planning. Their proposed batch policy is to pick the most-likely positive points (those farthest from an SVM hyperplane), which is quite different from our more-principled approach using Bayesian decision theory. Their policy is an analog of the one-step (greedy) myopic policy in our treatment, which performs poorly as we will show in Section 5.

Active search is a specific realization of active learning (AL). Though highly related, AL and AS have fundamentally different goals: learning an accurate model versus retrieving positive examples. One might argue that AS can be reduced to AL by first learning the decision boundary, then just collecting the predicted positive examples. However, it is often the case that the given budget is far from enough for an accurate model to be learned, and we must have more-elegant approaches to balance exploration and exploitation. Good AL policies could perform poorly in AS: Warmuth et al. [25] compared several variants of uncertainty sampling (arguably one of the most popular AL policies) with greedy AS policies, and demonstrated that the greedy policies performed much better in terms of retrieving active compounds. Jiang et al. [12] showed a $k$-nn classification model trained on 10 times more data still retrieved significantly fewer positives than a simple greedy AS policy. In their supplementary materials, they also showed that uncertainty sampling performed much worse than the greedy policy given the same budget.

Batch policies have been studied extensively in active learning [2–4, 11]. In particular, Chen and Krause [3] proposed an adaptive submodular objective function, and chose points greedily by maximizing the marginal gain. This algorithm is similar in spirit to our batch-ENS policy, though it is not known whether the batch-ENS function is submodular. They also proved a result similar in spirit to our Theorem 1 (also called "adaptivity gap" in [1]) to show that the price of parallelism is bounded irrespective of batch sizes. This theorem holds under the stochastic submodular maximization setting where the outcomes of variables are independent, which certainly does not apply in our case.

AS can be considered as Bayesian optimization (BO) with binary observations and cumulative reward maximization on a finite domain. Numerous batch BO policies have been studied [8–10, 28] Ginsbourger et al. [8, 9] proposed $q$-EI, in which $q$ points are selected simultaneously to maximize the expected improvement. They also used sequential simulation to optimize the $q$-EI objective, and proposed two heuristic "fictional oracles" called the Kriging believer (KB) and constant liar (CL). KB sets the label of a chosen point to its posterior mean, and CL sets the label to be a chosen constant, such as the maximum, mean, or minimum of the observed values so far. This is similar to our pessimistic or optimistic oracles.

Active search is also related to the multi-armed bandit (MAB) setting if a point is considered an arm and each point can only be pulled once. In the Gaussian process (GP) bandit optimization setting, Desautels et al. [5] proposed GP-BUCB, a batch extension of the GP-UCB policy [19]. They also construct the batch by sequentially simulating the GP-UCB policy, where the values of the selected points are "hallucinated" with the posterior mean, equivalent to the Kriging believer heuristic for $q$-EI. A similar strategy was adopted also in [27] to identify the compounds with the top-$k$ continuous-valued binding activities against an identified biological target. These approaches don't directly apply to our setting, where the target values are binary. In fact, Jiang et al. [12] showed that a UCB-style policy adapted to the Bernoulli setting performs worse than a myopic two-step policy on a range of problems in their supplementary material.

Table 1: Results for drug discovery data: Average number of positive compounds found by the baseline *uncertain-greedy* batch, greedy-batch, sequential simulation and batch-ENS policies. Each column corresponds to a batch size, and each row a policy. Each entry is an average over 200 experiments (10 datasets by 20 experiments). The budget $T$ is 500. Highlighted are the best (bold) for each batch size and those that are not significantly worse (blue italic) than the best under one-sided paired $t$-tests with significance level $\alpha = 0.05$.

|  | 1 | 5 | 10 | 15 | 20 | 25 | 50 | 75 | 100 |
|---|---|---|---|---|---|---|---|---|---|
| UGB | - | 257.6 | 257.9 | 258.3 | 250.1 | 246.0 | 218.8 | 206.2 | 172.1 |
| greedy | 269.8 | 268.1 | 264.1 | 261.6 | 258.2 | 257.0 | 240.1 | 227.2 | 208.2 |
| ss-one-1 | 269.8 | 260.7 | 254.6 | 245.2 | 233.6 | 223.4 | 200.8 | 182.9 | 178.9 |
| ss-one-m | 269.8 | 264.5 | 257.7 | 250.0 | 244.4 | 236.5 | 211.7 | 195.4 | 179.4 |
| ss-one-s | 269.8 | 266.8 | 261.3 | 256.7 | 248.7 | 244.1 | 214.9 | 202.4 | 181.3 |
| ss-one-0 | 269.8 | 268.1 | 264.1 | 261.6 | 258.2 | 257.0 | 240.1 | 227.2 | 208.2 |
| ss-two-1 | 281.1 | 237.1 | 219.8 | 210.8 | 212.1 | 196.2 | 172.1 | 158.8 | 152.9 |
| ss-two-m | 281.1 | 252.6 | 246.4 | 237.2 | 232.9 | 225.1 | 200.2 | 181.6 | 167.2 |
| ss-two-s | 281.1 | 248.9 | 242.5 | 235.3 | 226.6 | 219.2 | 196.7 | 175.3 | 158.3 |
| ss-two-0 | 281.1 | 252.5 | 247.6 | 247.9 | 244.4 | 240.4 | 225.6 | 213.8 | 199.1 |
| ss-ENS-1 | **295.1** | 269.4 | 247.9 | 227.2 | 223.1 | 210.3 | 185.3 | 152.6 | 148.7 |
| ss-ENS-m | *295.1* | 293.8 | 290.2 | 285.3 | 281.6 | 274.4 | 249.4 | 217.2 | 203.1 |
| ss-ENS-s | *295.1* | 289.9 | 278.3 | 269.8 | 262.6 | 255.0 | 220.8 | 185.5 | 161.2 |
| ss-ENS-0 | *295.1* | 293.6 | 289.1 | 288.1 | *287.5* | 280.7 | 269.2 | 257.2 | 241.0 |
| batch-ENS-16 | *295.1* | **300.8** | **296.2** | 293.9 | **292.1** | *288.0* | 275.8 | *272.3* | 252.9 |
| batch-ENS-32 | *295.1* | *300.8* | *295.5* | **297.9** | *290.6* | **288.8** | **281.4** | **275.5** | **263.5** |

# 5 Experiments

In this section, we comprehensively compare all our 14 proposed policies: (1) greedy-batch, coded as "greedy"; (2–13) sequential simulation, coded as "ss-P-O", where P (for policy) could be "one" (for one-step), "two" (for two-step), or "ENS", and O (for oracle) could be "s" (sampling), "m" (most-likely), "0" (pessimistic, i.e., always-0), or "1" (optimistic, i.e., always-1); (14) batch-ENS. Suggested by one of the the anonymous reviewers, we also compare these policies against another naïve baseline, which we call *uncertain-greedy* batch (UGB), where we build batches that simultaneously encourage exploration and exploitation by combining the most uncertain points and the highest probability points. We use a hyperparamter $r \in (0, 1)$ to control the proportion, choosing the most uncertain points for $100r\%$ of the batch, and greedy points for the remaining $100(1 - r)\%$ of the batch. We run this policy for $r \in \{0.1, 0.2, \ldots, 0.9\}$, and show the best result among them. We implement all these policies with the MATLAB active learning toolbox.[1] Following Jiang et al. [12], we consider data from three application domains: a citation network, material science, and drug discovery. Similar patterns could be found on the three domains, so we mainly present the results for 10 drug discovery datasets in the main text. Results for other datasets are detailed in the supplemental material. We use $k$ nearest neighbor ($k$-nn) with $k = 100$ as our probability model for the drug discovery datasets, and $k = 50$ for the other two datasets (following the studies in [7, 12]).

## 5.1 Drug discovery

We conduct our main investigation on a drug discovery application. In this application, our goal is to find chemical compounds that exhibit binding activity with a target protein. Each target protein defines an active search problem. We consider the first ten of the 120 datasets used in [7, 12] and only the ECFP4 fingerprint, which showed the best performance in those studies. These datasets share a pool of 100 000 negative compounds randomly selected from the ZINC database [20]. The number of positives of the ten datasets varies from 221 to 1024, with mean 553.

For each dataset, we start with one random initial positive seed observation and repeat the experiment 20 times. We test for batch sizes $b \in \{5, 10, 15, 20, 25, 50, 75, 100\}$, we also show the results for sequential search ($b = 1$) as a reference. The budget is set as $T = 500$. We test batch-ENS with

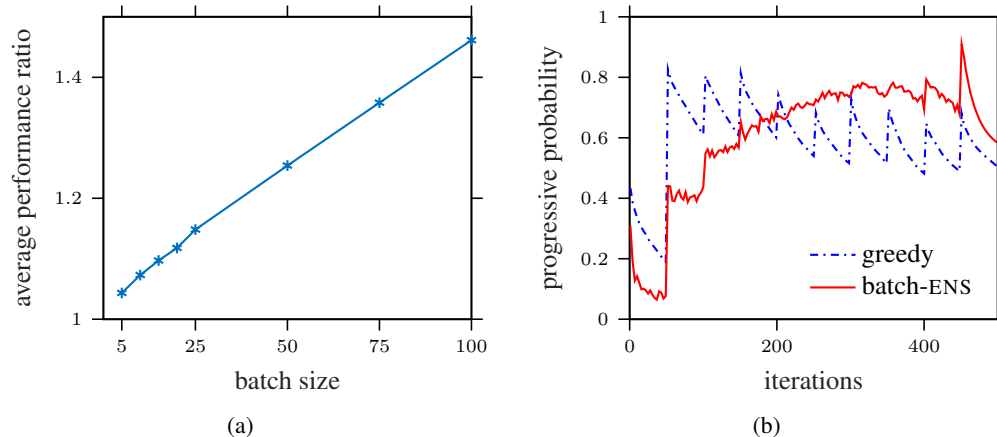

Figure 1: (a) Average performance ratio between sequential policies and batch policies, as a function of batch size, produced using averaged results in Table 1. (b) Progressive probabilities of the chosen points of greedy and batch-ENS-32, averaged over results for batch size 50 on all 10 drug discovery datasets and 20 experiments each.

16 and 32 samples, coded as batch-ENS-16 and batch-ENS-32. We show the number of positive compounds found in Table 1, averaged over the 10 datasets and 20 experiments each, so each entry in the table is an average over 200 experiments. We highlight the best result for each batch size in boldface. We conduct a paired $t$-test for each other policy against the best one, and also emphasize those that are not significantly worse than the best with significance level $\alpha = 0.05$ in blue italics.

We highlight the following observations. (1) The performance decreases as the batch size increases. (2) Nonmyopic policies are consistently better than myopics ones; in particular, batch-ENS is a clear winner. (3) For sequential simulation policies, the pessimistic oracle is almost always the best.

For batch-ENS, we find batch-ENS with 32 samples often performs better than with 16, especially for larger batch sizes. We have run batch-ENS for $b = 50$ with $N \in \{2, 4, 8, 16, 32, 64\}$ (see supplemental material), and find that the performance improves considerably as the number of samples increases, but the magnitude of this improvement tends to decrease with larger numbers. We believe 32 label samples offers a good tradeoff between efficiency and accuracy for $b = 50$.

## 5.2 Discussion

We now discuss our observations in more detail. First we see all our proposed policies perform better than the heuristic uncertain-greedy batch, even if we optimistically assume the best hyperparameter of this policy (not to mention we hardly know what the best hyperparameter should be in practice). Our framework based on Bayesian decision theory offers a more principled approach to batch active search (especially batch-ENS); and our methods are effectively hyperparameter-free (except the number of samples used in batch-ENS). In the following, we elaborate on the three observations.

**Empirical adaptivity gap.** Regardless of what policy is used, the performance in general degrades as the batch size increases. But how fast? We average the results in Table 1 over all policies for each batch size $b$ as an empirical surrogate for $\text{OPT}_b$ in Theorem 1, and plot the resulting surrogate value of $\frac{\text{OPT}_1}{\text{OPT}_b}$ as a function of $b$ in Figure 1a. Although these policies are not optimal, the empirical performance gap matches our theoretical linear bound surprisingly well. Similar results for different budgets on a different dataset are shown in the supplemental material. These results could provide valuable guidance on choosing batch sizes.

Despite the overall trends in our results, we see some interesting exceptions. That is, batch-ENS with batch size 5 is significantly better than that with batch size 1, with a $p$-value of $0.02$ under a one-sided paired $t$-test. This is counterintuitive based on our analysis regarding the adaptivity gap. We conjecture that batch-ENS with larger batch sizes forces more (but not too much) exploration, potentially improving somewhat on sequential ENS in practice.

**Why is the pessimistic oracle better?** Among the four fictional oracles, the pessimistic one usually performs the best for sequential simulation. When combined with a greedy policy, we have provided some mathematical rationale in Proposition 1: sequential simulation then near-optimally maximizes the probability of unit improvement, which is a reasonable criterion. Intuitively, by always assuming the previously added points to be negative, the probabilities of nearby points are lowered, offering a repulsive force compelling later points to be located elsewhere, leading to a more diverse batch. This mechanism could help better explore the search space.

This hypothesis is verified by quantifying the diversity of the chosen batches. Specifically, for each batch $\mathcal{B}$, for any $x_i, x_j \in \mathcal{B}$, we compute the rank of $x_j$ according to increasing distance to $x_i$, and average the ranks for all pairs as the diversity score of this batch. We use rank instead of distance for invariance to scale. We find that the diversity scores of the chosen batches align perfectly well with batch active search performance. Details can be found in the supplemental material. Note this coincides with the idea of explicitly using repulsion to create a diverse batch, which has been adopted in similar settings such as Bayesian optimization [10].

**Myopic vs. nonmyopic behavior.** Nonmyopic policies (ENS-based) almost always perform better than myopic policies. This certainly matches our expectation as nonmyopic policies are always cognizant of the budget and hence can better trade off exploration and exploitation [12]. To gain some insight into the nature of this myopic/nonmyopic behavior, in Figure 1b we plot the probabilities of the points chosen (at the iteration of being chosen) by the greedy and batch-ENS-32 policies for batch size 50 across the drug discovery datasets. Corresponding plots for other policies are shown in the supplemental material. First, in each batch, the trend for greedy is not surprising, since every batch represents the top-50 points ordered by probabilities. For batch-ENS, there is no such trend except in the last batch, where batch-ENS naturally degenerates to greedy behavior. Second, along the whole search process, greedy has a decreasing trend, likely due to over-exploitation in early stages. On the other hand, batch-ENS has an increasing trend. This could be partly due to more and more positives being found. More importantly, we believe this trend is in part a reflection of the nonmyopia of batch-ENS: in early stages, it tends to explore the search space, so low probability points might be chosen. As the remaining budget diminishes, it becomes more exploitive; in particular, the last batch is purely exploitive.

# 6 Conclusion

We have completed the first study on batch active search, where the goal is to find as many positives as possible in a given labeling budget. We derived the Bayesian optimal policy for batch active search, and proved a lower bound, linear in batch size, on the performance gap between optimal sequential and batch policies. This was shown to match empirical results.

We then generalized a recently proposed efficient nonmyopic search (ENS) policy to the batch setting and proposed two approaches to approximately solving the batch version of ENS: sequential simulation with fictional labeling oracles and greedy set function maximization. We conducted comprehensive experiments on data from three application domains evaluating all fourteen proposed policies. Results show that nonmyopic policies perform significantly better than myopic ones. By analyzing the results, we gained a deeper understanding of the nonmyopic behavior and find diversity to be an importantact consideration for batch policy design. We believe our theoretical and emprical analysis constitute a valuable step towards more-effective application of (batch) active search in various important domains such as drug discovery and materials science.

## Acknowledgments

We would like to thank all the anonymous reviewers for valuable feedbacks. SJ, GM, and RG were supported by the National Science Foundation (NSF) under award number IIA–1355406. GM was also supported by the Brazilian Federal Agency for Support and Evaluation of Graduate Education (CAPES). MA was supported by NSF under award number CNS–1560191. BM was supported by a Google Research Award and by NSF under awards CCF–1830711, CCF–1824303, and CCF–1733873.

## Footnotes

[1] https://github.com/rmgarnett/active_learning

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
