[Supplementary Material]

# Supplementary materials for efficient nonmyopic batch active search

**Shali Jiang**
CSE, WUSTL
St. Louis, MO 63130
jiang.s@wustl.edu

**Gustavo Malkomes**
CSE, WUSTL
St. Louis, MO 63130
luizgustavo@wustl.edu

**Matthew Abbott**
CSE, WUSTL
St. Louis, MO 63130
mbabbott@wustl.edu

**Benjamin Moseley**
Tepper School of Business, CMU and
Relational AI
Pittsburgh, PA 15213
moseleyb@andrew.cmu.edu

**Roman Garnett**
CSE, WUSTL
St. Louis, MO 63130
garnett@wustl.edu

## 1 Proof of Theorem 1

We have the following theorem in the main text:

**Theorem 1.** *There exist active search instances with budget $T$, such that*

$$\frac{\mathrm{OPT}_1}{\mathrm{OPT}_b} = \Omega\left(\frac{b}{\log T}\right),$$

*where $\mathrm{OPT}_x$ is the expected number of targets found by the optimal batch policy with batch size $x \geq 1$.*

*Proof.* We begin the proof by constructing a class of active search instances $\mathcal{I}$, parameterized by a given budget $T$ and batch size $b$, illustrated in Figure 1. In these instances, there are two types of points. Points of the first type are organized in a complete binary tree of height $h$, which is a parameter we will specify later. There are $2^h - 1$ such points, each with marginal probability $p$ of being positive, where $p$ is a also parameter we will fix later.

The second type of points is "clump points." There are $2^h$ "clumps" of points attached to the leaves of the binary tree, and each clump has size $T - h$. The labels in each clump are perfectly correlated; that is, either all labels in a clump are positive or all are negative. We construct the problem instance such that exactly one of these clumps contains positive points.

We denote the clumps as $\{C_j\}_{j=1}^{2^h}$, where each clump $C_j$ corresponds to a path $\mathcal{P}_j = D_1 D_2 \cdots D_h$ from the root of the binary tree to the clump, where $D_i \in \{\mathcal{L}, \mathcal{R}\}$ indicates progressing from a parent to its left ($\mathcal{L}$) or right ($\mathcal{R}$) child.

The positive clump can be identified by the following rule: start from the root of the binary tree and progress left if its label is negative and right if its label is positive. Repeat this procedure until a clump is reached. This clump is defined to be the positive clump, and we will refer to the single path leading to the positive clump as the *correct path*.

For example, in Figure 1, if the labels of the points on the red dashed path are 0, 1, and 0, respectively, then the third clump from the left would be the positive clump, and all others would contain negative points only.

To better understand the correlations among the labels of the tree nodes and the clumps, we define a notion of consistency between a labeling and a path.

**Definition 1.** *The labels $\{y_i\}_{i=1}^h$ of the nodes along a given path $\mathcal{P} = D_1 D_2 \cdots D_h$ corresponding to a clump $C$ are* consistent *with $\mathcal{P}$ if we have $y_i = 0$ when $D_i = \mathcal{L}$ and $y_i = 1$ when $D_i = \mathcal{R}$ for all labels along the path.*

There are $2^h$ possible labelings for the $h$ tree nodes on each path, but only one of them is consistent with the path. Among all the $2^h$ paths, the one with consistent labeling would correspond to the positive clump. Therefore, identifying the positive clump exactly specifies all the labels of the nodes on the correct path and also constrains all other clumps to contain negative points only.

However, identifying a clump as negative only implies that the joint probability of the consistent labeling of the nodes on its path is zero; but any other labeling of these points is still possible. The marginal probability of any point on this path does not change given this information unless there is only one point on this path remaining unobserved. The probability of its closest unobserved sibling clumps would also be increased given this information. For example, if the leftmost clump in Figure 1 is observed as being negative, this does not imply its immediate parent tree node is positive, but it does imply that the probability of its closest unobserved sibling clump would be increased by an appropriate factor. We will characterize this joint probability distribution more in Lemma 5.

Figure 1: An illustrative example of the constructed instance with $h = 3$ and $2^h = 8$ clumps, where the third clump from the left is positive, corresponding to the correct path, 010.

**Parameter settings.** We set the height of the tree $h = T/2$, and hence the clump size is also $T/2$. We set $p = \frac{100 \log T}{b}$, and here we assume $b$ is of higher order than $\log T$.

**Lower bound the optimal sequential policy.**

**Lemma 1.** $\text{OPT}_1 > T/2$.

*Proof.* Consider the following sequential policy. Query the tree along a path from the top down, selecting each point based on the label of the parent to maintain consistency. This policies identifies the positive clump in $h = T/2$ steps. The remaining budget, $T/2$, can then be spent querying the points in the positive clump. The expected performance of this policy certainly lower bounds that of the optimal one. Hence

$$\text{OPT[SAS]} \geq T/2 \cdot p + T/2 > T/2. \tag{1}$$

$\square$

**Upper bound the optimal batch policy.**

**Lemma 2.** $\text{OPT}_b = \mathcal{O}\left(\frac{T \log T}{b}\right)$.

First we claim that an optimal batch policy should never alternately query tree points and clump points before the positive clump is identified. Let the trace of an optimal batch policy be $B_1, B_2, \ldots, B_t$, where $t = T/b$, $B_i$ is the $i$th batch. Assume the positive clump is identified at the $k$th iteration; if it is never identified, let $k = t+1$. We claim that $B_1, \ldots, B_{k-1}$ should never alternate between tree points and clump points. That is, if $B_j$ is the first batch that contains clump points, then $B_{j+1}, \ldots, B_{k-1}$ would only contain clump points; otherwise, say $B_{j'} (j < j' \leq k - 1)$ contains tree points, then it's

only better to move the tree points in $B_{j'}$ to the position of $B_j$. This is because the immediate utility of querying tree points is always[1] the same before the positive clump is identified, no matter earlier or later; but earlier can only result in higher expected future utility for clump points. Once the positive clump is identified ($k \leq t$), then after $B_k$, we know all other clumps are negative, and we also know the labels of all the nodes on path corresponding to the positive clump, and all remaining tree points become totally independent. Hence the optimal batches $B_{k+1}, B_{k+2}, \ldots, B_t$ would be just collecting all the known positives; if there is still budget remaining after that, query the tree points randomly (no difference).

So there are only two possible cases for the optimal policy: (1) query tree points first, then clumps points; if the positive clump is identified, possibly query more tree points; (2) query clumps points only; if the positive clump is identified, possibly query some tree points.

Based on this observation, we upper bound the expected utility of the optimal batch policy by allowing budget $2T$, split by sub-policies $\mathcal{P}$ and $\mathcal{Q}$, where $\mathcal{P}$ only queries tree points with budget $T$, and $\mathcal{Q}$ only clump points also with budget $T$; whenever the positive clump is identified, we automatically grant utility $T$.

Let $u_1$ be the utility from tree nodes, $u_2$ be that from the clumps nodes. Let the expected utility of $\mathcal{P}$ be $\mathbb{E}_{\mathcal{P}}[u_1]$, and the expected utility of $\mathcal{Q}$ be $\mathbb{E}_{\mathcal{Q}}[u_2]$. We have

$$\text{OPT}[\text{BAS}] \leq \max_{\mathcal{P},\mathcal{Q}} \left\{ \mathbb{E}_{\mathcal{P}}[u_1] + \mathbb{E}_{\mathcal{P}}\big[\mathbb{E}_{\mathcal{Q}}[u_2]\big], \mathbb{E}_{\mathcal{Q}}[u_2] + \mathbb{E}_{\mathcal{Q}}\big[\mathbb{E}_{\mathcal{P}}[u_1]\big] \right\}. \tag{2}$$

We will prove Lemma 2 by upper bounding both cases. First we upper bound the easier case, where one performs $\mathcal{Q}$ first and then $\mathcal{P}$, then the harder case, first $\mathcal{P}$ then $\mathcal{Q}$.

When one performs $\mathcal{Q}$ first and then $\mathcal{P}$, we have the following:

**Lemma 3.** *For any $\mathcal{P}$ and $\mathcal{Q}$, $\mathbb{E}_{\mathcal{Q}}[u_2] + \mathbb{E}_{\mathcal{Q}}\big[\mathbb{E}_{\mathcal{P}}[u_1]\big] = \mathcal{O}\left(\frac{T \log T}{b}\right)$.*

To prove this lemma, we upper bound the two parts separately.

**Lemma 4.** *For any $\mathcal{Q}$, $\mathbb{E}_{\mathcal{Q}}[u_2] = o(1)$.*

To prove this bound, we prove the following more general lemma, which will be also used later.

**Lemma 5.** *Given a tree of height $h'$, a policy $\mathcal{Q}$ with budget $T$ that only queries clump points has expected utility upper bounded by $U \equiv \frac{1}{2}T(T+1)(1-p)^{h'-\log(T+1)-1}$. Furthermore, the probability of identifying the positive clump is upper bounded by $\mathcal{O}(\frac{1}{T})$ if $h' = h$.*

*Proof.* For a tree of height $h'$, the probabilities of the clumps are non-increasing from left to right: $(1-p)^{h'} \equiv p_{\max}, (1-p)^{h'-1}p, (1-p)^{h'-1}p, (1-p)^{h'-2}p^2, \ldots, (1-p)p^{h'-1}, p^{h'}$.

An optimal policy $\mathcal{Q}$ (possibly $b = 1$) only querying clump points goes like this: select points from some clumps; if any one turns out to be positive, then spend the remaining budget only querying that clump and the positives on the corresponding path (as mentioned in the beginning, identifying the positive clump also reveals the labels of all points on its corresponding path); if all selected points turn out to be negative, continue to try other clumps in the same way.

The expected utility of this process can be upper bounded via the following strategy. We first compute an upper bound $p^*$ of the posterior probability of any clump conditioned on $T$ negative observations, then compute the expected utility of this process as if all clumps were independent and of probabilities $p^*$.

We first derive the maximum possible probability $p^*$ after any $T$ negative observations. For clumps under a subtree of height $k$, the maximum increase of probability is at most by a factor of $1/(1-p)^k$ by eliminating all $2^k - 1$ clumps except the leftmost one. In terms of increasing the maximum posterior probability by the most amount, the best strategy is to focus on a minimum subtree covering at least $T + 1$ clumps, and spend all the budget $T$ eliminating all other clumps ($2^k - 1$) except the leftmost

one. Let $k$ be the minimum integer such that $T \leq 2^k - 1$, we get $k = \lceil \log(T+1) \rceil \leq \log(T+1) + 1$. So observing any $T$ clumps (if no hit) can increase the probability of any other clump to at most $(1-p)^{h'}/(1-p)^k \leq (1-p)^{h'-\log(T+1)-1} \equiv p^*$. So the expected utility of any "clumps-only" policy can be upper bounded by (for any $b \geq 1$)

$$p^*T + p^*(T-b) + p^*(T-2b) + \cdots + p^*b$$
$$\leq \frac{1}{2}T(T+1)p^*$$
$$\leq \frac{1}{2}T(T+1)(1-p)^{h'-\log(T+1)-1} \equiv U.$$

The probability of identifying the positive clump $\Pr(hit)$ can also be upper bounded using $p^*$:

$$\Pr(\text{hit}) < 1 - (1-p^*)^T. \tag{3}$$

One can show that

$$\lim_{T \to \infty} \frac{1 - (1-p^*)^T}{1/T} = 0. \tag{4}$$

So $\Pr(\text{hit}) = o(1/T)$.

$\square$

*Proof of Lemma 4.* Using Lemma 5, we can bound $\mathbb{E}_\mathcal{Q}[u_2] \leq U$ with $h' = h$, and it's easy to derive $U$ is $o(1)$ in this case. $\square$

**Lemma 6.** *For any* $\mathcal{P}, \mathcal{Q}$, $\mathbb{E}_\mathcal{Q}\big[\mathbb{E}_\mathcal{P}[u_1]\big] = \mathcal{O}\left(\frac{T \log T}{b}\right)$.

*Proof.* Consider two cases after $\mathcal{Q}$ finishes:

- First, $\mathcal{Q}$ identified the positive clump, with probability at most $\mathcal{O}(1/T)$ by Lemma 5. In this case the correct path would also be identified, and there are at most $h$ positives on the path. We can simply upper bound the utility by $T$. So $\mathbb{E}_\mathcal{P}[u_1] \leq \frac{1}{T} \cdot T = 1$;

- Second, $\mathcal{Q}$ did not identify the positive clump, in which case each tree point almost always has expected utility $p$. A subtle situation is when all but one point are queried on a path corresponding to a known negative clump, in which case the label of this point would be known already. So it's possible to get utility 1 instead of $p$ when querying a tree point. However, this happens with extremely small probability (at most $(1-p)^{T/2-1}$), since all but one point on the path (length $T/2$) must have consistent labels with the path. So we can simply ignore the expected utility of this case. Hence $\mathbb{E}_\mathcal{P}[u_1] \leq Tp = \mathcal{O}\left(\frac{T \log T}{b}\right)$.

If we sum these two cases, we have $\mathbb{E}_\mathcal{Q}\big[\mathbb{E}_\mathcal{P}[u_1]\big] = \mathcal{O}\left(\frac{T \log T}{b}\right)$.

$\square$

*Proof of Lemma 3.* By Lemma 4 and 6, we have $\mathbb{E}_\mathcal{Q}[u_2] + \mathbb{E}_\mathcal{Q}\big[\mathbb{E}_\mathcal{P}[u_1]\big] = \mathcal{O}\left(\frac{T \log T}{b}\right)$. $\square$

Now consider the first case where one performs $\mathcal{P}$ first and then $\mathcal{Q}$, we also have

**Lemma 7.** *For any* $\mathcal{P}, \mathcal{Q}$, $\mathbb{E}_\mathcal{P}[u_1] + \mathbb{E}_\mathcal{P}\big[\mathbb{E}_\mathcal{Q}[u_2]\big] = \mathcal{O}\left(\frac{T \log T}{b}\right)$.

We also upper bound the two parts separately.

**Lemma 8.** *For any* $\mathcal{P}$, $\mathbb{E}_\mathcal{P}[u_1] = \mathcal{O}\left(\frac{T \log T}{b}\right)$.

Figure 2: Illustration of the relevant subtree. The red curve shows the correct path identified so far. If the last node on the correct path is negative, then the node $A$ must also be on the correct path, and the subtree rooted at $A$ is the relevant subtree; otherwise $B$ is on the correct path, and the subtree rooted at $B$ is the relevant subtree.

*Proof.* This is trivial to show due to independence of the nodes: $\mathbb{E}_{\mathcal{P}}[u_1] \leq Tp = T \cdot \frac{100 \log T}{b} = \mathcal{O}\left(\frac{T \log T}{b}\right)$. $\square$

**Lemma 9.** *For any* $\mathcal{P}, \mathcal{Q}$, $\mathbb{E}_{\mathcal{P}}\big[\mathbb{E}_{\mathcal{Q}}[u_2]\big] = \mathcal{O}(1)$.

Note a key property of the constructed instance is that one has to wait until the previouly chosen points are observed to determine which direction to go. However, in BAS, a batch of points has to be observed simultaneously, hence in each batch, various number of observations could be wasted depending on the how many points turn out to be on the correct path. This is exactly why there could be a gap between the performances of SAS and BAS. In the following, we will upper bound the expected performance in this case by bounding the probability of reaching a certain level on the correct path after all $T/b$ batch queries. We first introduce some useful definitions:

**Definition 2.** *For any observed node in the tree, if it is positive, then we call its right subtree* relevant*, and its left subtree* irrelavant*; or the other way around if it is negative.*

**Definition 3.** *The* relevant subtree *at any step is acquired by trimming all observed nodes and their irrelevant subtrees. That is, removing any observed node and its irrelevant subtree, and reconnecting its parent to its relevant subtree.*

Let $P^* = \arg\max_{\mathcal{P}} E_{\mathcal{P}}\big[\max \mathbb{E}_{\mathcal{Q}}[u_2]\big]$ be the optimal sub-policy querying the tree. It's easy to see that $P^*$ should always only query points in the relevant tree, since querying anywhere else has same $u_1$ but reveals no information about the identity of the positive clump.

**Definition 4.** *A sequence $S$ of length $\ell$ is a set of $\ell$ points $x_1, \ldots, x_\ell$ in a relevant subtree such that $x_i$ is an ancestor of $x_{i+1}$ for all $i = 1, \ldots, \ell - 1$.*

More intuitively, a set of points is a sequence if they are contained in some path.

*Proof of Lemma 9.* Recall $p = \frac{100 \log T}{b} \to 0$ as $T \to \infty$ since we assumed $b$ is of higher order than $\log T$, so $p$ is close to 0 (hence $p \ll 1/2$) for large enough $T$.

For any sequence $S$ of length $\ell$, the probability of $S$ lying on the correct path is upper bounded by

$$(1-p)^\ell = \left[\left(1 - \frac{1}{1/p}\right)^{-1/p}\right]^{-p\ell} \approx \exp(-p\ell) = \exp\left(-\frac{100 \log T}{b}\frac{b}{10}\right) = \frac{1}{T^{10}}.$$

Given any batch of size $b$, there are at most $b$ sequences of length $\ell$. So the probability that this batch contains $\ell$ nodes on the correct path can be union-bounded by

$$\frac{1}{T^{10}}b \leq \frac{1}{T^{10}}T = \frac{1}{T^9}.$$

The probability of any $T/b$ batches contains $\frac{T}{b} \cdot \frac{b}{10} = \frac{T}{10}$ nodes on the correct path can be union-bounded by

$$\frac{1}{T^9} \cdot \frac{T}{b} = \frac{1}{T^8 b} \leq \frac{1}{T^8}. \tag{5}$$

For any policy (e.g., $\mathcal{P}^*$) iteratively choosing a set $A$ of $T$ points with batch size $b$, two cases can happen:

- case 1: $A$ contains $T/10$ or more points on the correct path. We have shown the probability of this happening is at most $1/T^8$. In this case we simply upper bound $\mathbb{E}_{\mathcal{Q}}[u_2]$ by $T$.

- case 2: $A$ contains less than $T/10$ points on the correct path. Then the remaining unidentified points on the correct path is at least $h' = T/2 - T/10 = 2T/5$. That is, the shallowest leaf in the relevant subtree has height at least $h'$. The distribution of the remaining clumps is dominated by that of a brand new active search instance $\mathcal{A}$ with the tree height $h'$, in the sense that the optimal expected utility $u_2'$ of $\mathcal{Q}$ on $\mathcal{A}$ can only be greater, that is, $\mathbb{E}_{\mathcal{Q}}[u_2] \leq \mathbb{E}_{\mathcal{Q}}[u_2']$. Applying Lemma 5 with $h' = 2T/5$, we can find $\mathbb{E}_{\mathcal{Q}}[u_2'] = o(1)$.

Therefore, there exist a constant $c$ and large enough $T$, such that

$$\mathbb{E}_{\mathcal{P}}\big[\mathbb{E}_{\mathcal{Q}}[u_2]\big] \leq \Pr(\text{case 1}) \cdot T + \Pr(\text{case 2}) \cdot c$$
$$\leq \frac{1}{T^8} \cdot T + 1 \cdot c,$$

which is $\mathcal{O}(1)$. □

*Proof of Lemma 7.* Combining Lemma 8 and 9, we have $\mathbb{E}_{\mathcal{P}}[u_1] + \mathbb{E}_{\mathcal{P}}\big[\mathbb{E}_{\mathcal{Q}}[u_2]\big] = \mathcal{O}\left(\frac{T \log T}{b}\right)$. □

*Proof of Lemma 2.* By (2) and Lemma 3 and 7, we have

$$\text{OPT}_b = \mathcal{O}\left(\frac{T \log T}{b}\right). \tag{6}$$

□

Combining Lemma 1 and Lemma 2, we have

$$\frac{\text{OPT}_1}{\text{OPT}_b} = \Omega\left(\frac{b}{\log T}\right).$$

□

## 2 Proof of Proposition 1

We had the following proposition in the main text:

*Proposition 1: The batch constructed by sequentially simulating one-step Bayesian optimal policy with pessimistic fictional oracle is near-optimally maximizing the probability that at least one of the points in the batch is positive, assuming the probability model $\mathcal{P}$ is such that observing more negative points does not increase the probability of any other point being positive.*

*Proof.* The probability of a batch $\mathcal{B} = \{x_1, x_2, \ldots, x_b\}$ having at least one positive can be written as

$$g(\mathcal{B}) = 1 - \Pr(y_1 = 0 \wedge y_2 = 0 \wedge \cdots \wedge y_b = 0). \tag{7}$$

It's easy to see $g(\emptyset) = 0$. It's also easy to see $g(\mathcal{B})$ is monotone. That is, for any $A \subseteq B \subseteq \mathcal{X}$, $g(A) \leq g(B)$. Now we show $g(\mathcal{B})$ is submodular. For any set $\mathcal{B}$, define $\mathcal{B} = 0$ as the event that $\forall x \in \mathcal{B}$, its label $y = 0$. The marginal gain of any $x \in \mathcal{X}$ (with label $y$) is

$$g(\mathcal{B} \cup \{x\}) - g(\mathcal{B})$$

$$
\begin{aligned}
&= \Pr(\mathcal{B} = 0) - \Pr(\mathcal{B} = 0 \wedge y = 0) \\
&= \Pr(\mathcal{B} = 0) - \Pr(y = 0 \mid \mathcal{B} = 0) \Pr(\mathcal{B} = 0) \\
&= \Pr(y = 1 \mid \mathcal{B} = 0) \Pr(\mathcal{B} = 0).
\end{aligned}
\tag{8}
$$

Let $A \subseteq B \subseteq \mathcal{X}$, $x \in \mathcal{X} \setminus B$ and its label $y$, we have

$$
\begin{aligned}
g(A \cup \{x\}) - g(A) &= \Pr(y = 1 \mid A = 0) \Pr(A = 0); \\
g(B \cup \{x\}) - g(B) &= \Pr(y = 1 \mid B = 0) \Pr(B = 0).
\end{aligned}
$$

Since $A \subseteq B$, we have $\Pr(B = 0) \leq \Pr(A = 0)$. We also have $\Pr(y = 1 \mid B = 0) \leq \Pr(y = 1 \mid A = 0)$ due to the (very reasonable) assumption that observing more negative points does not increase the probability of any other point being positive. Hence

$$
g(B \cup \{x\}) - g(B) \leq g(A \cup \{x\}) - g(A).
\tag{9}
$$

Therefore $g$ is a submodular function.

Observing (8), we see sequentially simulating one-step Bayesian optimal policy (i.e., choose the point with maximum probability) with a pessimistic (always-0) fictional oracle is exactly greedily maximizing the marginal gain of $g(\mathcal{B})$. By the classical results in [5], we know this greedy solution has approximation ratio $1 - 1/e \approx 0.6321$. $\qquad\square$

This proposition is inspired by Wang [7], applying Bayesian active learning to a biological application. The goal of Wang [7] was to find a peptide as short as possible that is substrate for two protein-modifying enzymes. They showed that under their setting, for any set of peptides $S$ and the currently known shortest length $\ell$, the probability of improvement (i.e., $S$ contains a positive peptide shorter than $\ell$) is a submodular function, and greedy maximization is equivalent to choosing from the peptides shorter than $\ell$ that has maximum probability of being positive, conditioned on all previously chosen ones being negative.

## 3  More Results

### 3.1  Finding NeurIPS papers in a citation network

In this experiment, we consider a subset of the CiteSeer$^x$ citation network (first described by Garnett et al. [2]) comprised of 39 788 computer science papers published in the top-50 most popular computer science venues. Among them there are 2 190 NeurIPS paper (5.5%), and our active search task is to find those NeurIPS papers. Note this is a challenging task since many very similar venues such as ICML, AAAI, IJCAI etc. are also in this network. We use experimental settings matching [3]. Specifically, we randomly select a single target (i.e., a NeurIPS paper) to form the initial observations $\mathcal{D}_0$, and repeat the experiment 100 times for each batch size and each policy. We set the budget $T$ to 500. We show the average number of NeurIPS papers found at termination for batch sizes $b \in \{5, 10, 15, 20, 25\}$ in Table 1. If $b$ does not divide $T$ (e.g., $b = 15$), we take $t = \lceil T/b \rceil$, and only count the first $T$ points. We also add the results for batch size 1 for reference. For batch-ENS, we use 16 samples; therefore for batch size 5 the expected future utility (Eq. (8) in the main text) is computed exactly. We highlight the best result for each batch size in boldface. We conduct a paired $t$-test [2] for each other policy against the best one, and also emphasize those that are not significantly worse than the best with significance level $\alpha = 0.05$ in blue italics. We use this convention in all tables.

We summarize similar patterns as results for drug discovery datasets described in the main text: (1) for this dataset the uncertain-greedy batch (UGB) is actually a strong baseline; it performed mostly better than the myopic policies. But keep in mind we are reporting the best result for UGB varying the hyperparameter; in practice, it is hard to know which hyperparameter is the best. However, it is still far worse than ss-ENS-0 and batch-ENS. (2) As a general trend, the performance decreases as batch size increases. (3) ss-ENS-0 and batch-ENS (both nonmyopic) are either the best or not significantly worse than the best for all batch sizes except 25. (4) Sequential simulation is almost always better with the pessimistic oracle, except with two-step for $b = 5$.

Table 1: Results for CiteSeer$^x$ data: Average number of targets found by various batch policies: greedy-batch, sequential simulation "ss-P-O" and batch-ENS, with batch sizes 5, 10, 15, 20, 25. The average is taken over 100 experiments. Highlighted are the best in each column and those not significantly worse than the best using a one-sided paired $t$ test with significance level $\alpha = 0.05$.

|          | 1     | 5     | 10    | 15    | 20    | 25    |
|----------|-------|-------|-------|-------|-------|-------|
| UGB      | -     | 159.2 | 153.5 | *155.5* | 149.3 | 147.9 |
| greedy   | 154.9 | 154.2 | 149.1 | 149.2 | 148.4 | *151.1* |
| ss-one-1 | -     | 152.1 | 141.7 | 126.2 | 121.4 | 117.9 |
| ss-one-m | -     | 152.4 | 141.7 | 132.0 | 127.8 | 127.4 |
| ss-one-s | -     | 149.2 | 147.7 | 146.7 | 142.5 | 132.8 |
| ss-one-0 | -     | 154.1 | 148.5 | 149.2 | 148.5 | *151.1* |
| ss-two-1 | 165.9 | 154.6 | 143.3 | 134.2 | 122.8 | 122.2 |
| ss-two-m | -     | 156.1 | 142.2 | 139.8 | 128.7 | 126.1 |
| ss-two-s | -     | 157.1 | 153.5 | 145.8 | 141.1 | 142.0 |
| ss-two-0 | -     | 156.6 | 154.7 | *152.4* | *151.6* | *154.1* |
| ss-ENS-1 | 187.2 | 154.7 | 142.6 | 135.0 | 131.9 | 122.1 |
| ss-ENS-m | -     | 154.2 | 143.0 | 138.1 | 135.7 | 126.6 |
| ss-ENS-s | -     | 165.5 | 155.8 | 147.5 | 142.8 | 139.3 |
| ss-ENS-0 | -     | *169.1* | **165.6** | *155.1* | *152.9* | 149.8 |
| batch-ENS | -    | **170.0** | *163.1* | **157.0** | **154.2** | **154.5** |

Table 2: Diversity scores of the chosen batches by all our proposed policies, measured by the average rank of distances from each other in a batch, produced from the results on CiteSeer$^x$ data. Higher value indicates more diversity.

|          | 5    | 10   | 15   | 20   | 25   |
|----------|------|------|------|------|------|
| greedy   | 1515 | 1970 | 2336 | 2443 | 2535 |
| ss-one-1 | 303  | 437  | 599  | 677  | 761  |
| ss-one-m | 866  | 810  | 1000 | 976  | 1111 |
| ss-one-s | 1221 | 1134 | 1458 | 1374 | 1692 |
| ss-one-0 | 1524 | 1983 | 2339 | 2450 | 2532 |
| ss-two-1 | 413  | 507  | 603  | 735  | 772  |
| ss-two-m | 927  | 982  | 987  | 1221 | 1322 |
| ss-two-s | 1344 | 1331 | 1467 | 1538 | 1483 |
| ss-two-0 | 1768 | 1933 | 2221 | 2540 | 2576 |
| ss-ens-1 | 968  | 1254 | 1328 | 1299 | 1400 |
| ss-ens-m | 1323 | 1458 | 1751 | 1683 | 1887 |
| ss-ens-s | 2131 | 2258 | 2370 | 2402 | 2602 |
| ss-ens-0 | 1987 | 2281 | 2542 | 2587 | 2725 |
| batch-ENS | 2266 | 2585 | 2842 | 3126 | 3225 |

## 3.2 Material Science Data: Finding Metallic Glasses

The goal here is to find novel alloys capable of forming metallic glasses (BMGs). Compared to crystalline alloys, BMGs have many desirable properties, including high toughness and good wear resistance. This dataset consists of 118 678 known alloys from the materials literature [4, 8], among which 4 746 (about 4%) are known to exhibit glass-forming ability, which we define as positive/targets. We conduct the same experiment as for CiteSeer$^x$ data. This dataset is much larger, so we only repeat the experiment 30 times, randomizing the initial seed. We also set $T = 500$. The results [3] are reported in Table 3.

We have the following observations: (1) The uncertain-greedy batch policy is again mostly worse than all our proposed batch active search policies. (2) The performance often decreases as batch size increases. But we see more exceptions than in the results for CiteSeer$^x$ data, probably due to fewer trials. (3) The nonmyopic policies (i.e., ENS based policies) consistently perform best for all batch sizes, and all myopic policies are significantly worse than the best policy. (4) Sequential simulation with the pessimistic oracle are mostly the best for one-step and two-step policies; however, we do not see the same pattern for ENS. At this moment we are not sure whether this is only because of lack of repetition or if this particular dataset has different properties.

Table 3: Results for BMG data: average number of targets found by various batch policies: baseline greedy-batch, sequential simulation "ss-P-O" and batch-ENS, with batch sizes 5, 10, 15, 20, 25. The average is taken over 30 experiments. Highlighted are the best in each column and those that are not significantly worse than the best using a one-sided paired $t$ test with significance level $\alpha = 0.05$.

|          | 1     | 5     | 10    | 15    | 20    | 25    |
|----------|-------|-------|-------|-------|-------|-------|
| UGB      | -     | 269.6 | 265.8 | 265.9 | 255.7 | 247.9 |
| greedy   | 283.7 | 278.1 | 272.5 | 269.6 | 264.2 | 263.6 |
| ss-one-1 | -     | 262.8 | 244.2 | 235.4 | 227.6 | 224.1 |
| ss-one-m | -     | 269.0 | 252.2 | 236.7 | 231.3 | 225.6 |
| ss-one-s | -     | 283.0 | 272.7 | 263.1 | 255.6 | 246.1 |
| ss-one-0 | -     | 278.1 | 272.5 | 269.8 | 264.3 | 263.6 |
| ss-two-1 | 282.0 | 270.7 | 242.7 | 241.4 | 231.4 | 225.6 |
| ss-two-m | -     | 272.5 | 250.9 | 243.6 | 242.5 | 226.4 |
| ss-two-s | -     | 274.4 | 267.8 | 262.3 | 250.9 | 251.6 |
| ss-two-0 | -     | 277.1 | 275.8 | 274.1 | 264.7 | 266.1 |
| ss-ENS-1 | 304.9 | *301.7* | **298.4** | 290.2 | 277.5 | 280.2 |
| ss-ENS-m | -     | **304.3** | *294.5* | 290.3 | 284.7 | 283.1 |
| ss-ENS-s | -     | 290.3 | 283.5 | 288.0 | **299.2** | *289.2* |
| ss-ENS-0 | -     | *301.4* | 281.7 | 283.8 | 279.5 | 275.7 |
| batch-ENS | -    | *300.6* | *296.2* | **306.7** | 287.6 | **294.8** |

## 3.3 Performance vs. Number of Samples for batch-ENS

We plot the performance on drug discovery datasets versus number of samples $N$ used for batch-ENS in Figure 3. The batch size here is 50. We can see usually doubling $N$ can bring significant improvement, except 16 over 8, and 64 over 32. We believe 32 is a good tradeoff of efficiency and accuracy for batch size 50. An interesting future work is theoretical analysis providing guidance for choosing $N$.

## 3.4 Diversity Scores

To verify our hypothesis that the pessimistic oracle performances better is due to encouraging diversity, we show in Table 2 the diversity scores of the chosen batches using the results on CiteSeer$^x$ data. The diversity is measured as follows: for each batch $\mathcal{B}$, for any $x_i, x_j \in \mathcal{B}$, we compute the rank of $x_j$ according to increasing distance to $x_i$, and average the ranks for all pairs as the diversity score of this batch. We use rank instead of distance for invariance to scale.

To better contrast, we extract the diversity scores for batch size 20, and present in Table 4. There is an increasing trend in each row, with always-0 the highest; also in each column, with ENS the highest.

If comparing Table 2 and Table 1 closely, one could find that the diversity scores and active search performances align remarkably well. This indicates diversity could be an important consideration for batch policy design.

Figure 3: Number of targets found versus number of samples used for batch-ENS. This is averaged over the results for batch size 50 on 10 drug discovery datasets and 20 experiments each. The text labels show the percentage of improvement and $p$-value of one-sided $t$ tests comparing against previous numbers, e.g., 8 samples improves over 4 samples by 1.9%, and the $p$-value is 0.04.

Table 4: Diversity scores of the chosen batches by all sequential simulation policies, measured by the average rank of distances from each other in a batch. The results are for $b = 20$ on CiteSeer$^x$ data. Higher values indicate more diversity. For reference, the score for greedy and batch-ENS are 2443 and 3126, respectively.

|      | 1    | m    | s    | 0    |
|------|------|------|------|------|
| one  | 677  | 976  | 1374 | 2450 |
| two  | 735  | 1221 | 1538 | 2540 |
| ENS  | 1299 | 1683 | 2402 | 2587 |

### 3.5  Behavior of the Policies

We plot the progressive probabilities of the chosen points (while being chosen) by all policies in Figure 4. We can see all myopic policies have similar pattern, and all nonmyopic policies also have similar pattern.

### 3.6  Pruning the search space

Jiang et al. [3] developed a pruning technique for ENS. The basic idea is: we first compute an upper bound $p^*(\mathcal{D})$ on marginal probabilities after observing $\mathcal{D}$ and one additional positive observation. We use this to compute an upper bound of the ENS score for each candidate point. We also compute the actual ENS score for the point with maximum probability, which serves as a global lower bound of the maximum score. Now any candidate point with upper bound less than the lower bound cannot possibly maximize the score and can be removed from consideration.

Here we further improve the pruning for ENS and generalize it to batch-ENS. First, instead of just computing a maximum probability bound $p^*(\mathcal{D})$, we can compute upper bounds of the highest $r \equiv T - b - |\mathcal{D}_i|$ probabilities $p_1^*(\mathcal{D}), p_2^*(\mathcal{D}), \ldots, p_r^*(\mathcal{D})$ after observing $\mathcal{D}$ and one additional positive. This generalization is trivial for a $k$-nn model. We can now upper bound the $\sum'_{T-b-|\mathcal{D}_i|}$ term more tightly by $\sum_{j=1}^{r} p_j^*(\mathcal{D}_i)$ than by $(T - b - |\mathcal{D}_i|)p^*(\mathcal{D}_i)$. This upper-bounding technique for ENS can be straightforwardly generalized to batch-ENS, where we only need to apply this upper bound for each sample (Eq. (8) in the main text) and average them.

Besides improving the upper bound, we can also improve the lower bound. As we continue to compute scores for more points, we can update the lower bound when observing a larger score. Tightening the bound enables more-aggressive pruning. To better use this improving lower bound, we first sort the candidates points in decreasing order of their upper bounds, and compute actual ENS

Figure 4: Progressive probabilities of the chosen points of all policies, averaged over results for batch size 50 on all 10 drug discovery datasets and 20 experiments each.

Figure 5: Illustration of pruning. The $x$-axis is the index of candidate points in descending order of the upper bounds, and the $y$-axis is the actual marginal gain of batch-ENS score as in Eq. (6) in the main text. These plots are generated from running batch-ENS on the CiteSeer$^x$ data with budget $T = 500$ and batch size $b = 5$. There are $39\,788$ points, we only plot the first and last $1\,000$ points to have a better presentation. (a) At the time of choosing the first point ($k = 1$) of the 99th batch (i.e., $i = 98$) when $T - b - |\mathcal{D}_i| = 5$; here 99.61% of points are pruned. (b) At the time of choosing the first point ($k = 1$) of the 16th (i.e., $i = 15$) batch when $T - b - |\mathcal{D}_i| = 420$; 98.23% of the points are pruned.

Figure 6: Average performance ratio between sequential policies and batch policies, as a function of batch size, produced using averaged results (excluding uncertain-greedy) for the CiteSeer dataset ( Table 1).

scores in this order. This way, we never unnecessarily compute scores for points that might be pruned later. This is similar to the idea of *lazy evaluation* [1] for efficiently maximizing the GP-UCB [6] score.

Figure 7c illustrates the pruning in a representative iterations of batch-ENS on CiteSeer$^x$ data with budget $T = 500$ and batch size $b = 5$. We see when the remaining budget is small ( $T - b - |\mathcal{D}_i| = 5$), the upper bound is extremely tight, and in this example 99.61% of the points are pruned.

We give another illustration of pruning when the remaining budget ($T - b - |\mathcal{D}_i|$) is large in Figure 5b. We can see in this case the upper bound is much looser, but we are still able to prune 98.23% of the points.

We present the results of pruning for all three types of datasets in Table 5. On all three types of datasets, the majority of points in each iteration are pruned. Especially on drug discovery datasets, on average over 98% of the points are pruned; that is over 50 times speed-up.

Table 5: Results for pruning effectiveness. The numbers are averaged over all iterations of batch-ENS for all batch sizes tested. For drug discovery data, the result is averaged over batch-ENS-16 and batch-ENS-32.

| datasets | #total | #pruned | pruning rate |
|---|---|---|---|
| CiteSeer$^x$ | 39546 | 27422.1 | 68.37% |
| BMG | 111360 | 82343.6 | 73.94% |
| Drug discovery | 100316 | 98612.6 | 98.30% |

### 3.7 Empirical adaptivity gap

In the main text, we showed that the performance ratio between batch and squential active search follows a linear trend, which is consistent to our Theorem 1. To verify the empirical linear trend also holds for different budgets and on other datasets, we plot the same curves for the CiteSeer dataset for budget $T = 100, 300, 500$ in Figure 6. We can see these lines all appear to be linear; more interestingly, given the same batch size, the gap is smaller for larger $T$, which is also indicated by Theorem 1. As future work, we will further investiage whether this gap will decrease as $1/\log T$.

### 3.8 Naive exploration/exploitation approaches

Active search is a special paradigm of active learning where the key chanllenge is to carefully balance exploration (learning) and exploitation (search). If we have a fully learned model, then active search

Figure 7: Comparion of active search policies with a naive exploration-exploitation approach called uncertain-then-greedy (UTG). (a) CiteSeer Dataset. (b) BMGs dataset. (c) Drug discovery datasets.

is trivial, since we only need to retrieve the points with highest probabilities. In practice, the budget might never be enough for a model to be fully learned, that's why we need to carefully allocate the budget for learning and search. One naive approach is to first spend some budget for learning, then use the remaining budget for search. Existing results [3] have shown that this naive approach is much worse than active search policies even granted 10 times more budget just for learning. Here we conduct more experiments in this regard motivated by the reviewers suggestions.

First, we implemented a strategy proposed by one of the reviewers, which we call *uncertain-then-greedy* (UTG). We first perform uncertainty sampling (arguably the most-popular active learning method) for a portion of the budget, then switch to greedy search. The transition point is controlled by a hyperparameter $r \in (0, 1)$: the first $100r\%$ of the budget is used for active learning (exploration), and the remaining $100(1 - r)\%$ is used for exploitation. We ran this policy on all three types of datasets described previously for $r \in \{0.1, 0.2, \ldots, 0.9\}$, with a batch size of 1 and budget $T = 500$. We repeated this experiments the same number of times with the same initial random seeds used for the other policies. The results are plotted in Figure 7, comparing with one- and two-step lookahead and ENS.

We can see that on the CiteSeer dataset this naïve approach indeed beats the greedy one-step lookahead policy, and approaches the performance of two-step lookahead when $r = 0.8$, but performs far worse than ENS; on the other two datasets, UTG does not show any advantage. Although we could probably strengthen this baseline with more-advanced active learning or active search policies, it will never

be clear how to best choose the transition hyperparameter $r$. Our approach, on the other hand, automatically transitions from exploration to exploitation in line with the optimal policy.

## Footnotes

[1]Except when the tree point is the only unobserved one along an otherwise consistent path corresponding to a clump known to be negative; this happens with extremely small probability as we will also see in Lemma 6.

[2]Wilcoxon signed rank tests give similar results.

[3]Note compared to results reported in [3], here the number of positives found is much smaller. This is because we excluded the 10% of training data for feature selection, whereas [3] did not. Please contact the authors for more details if interested.