[Reviews · NeurIPS 2018]

Reviewer 1



The submission generalizes another algorithm for non-myopic active search from an ICML 2017 paper. The submission proves a theoretical guarantee on the number of targets found with a batch policy versus online. The generalization proposed in the paper is fairly straightforward, and the sequential simulation and greedy approximation strategies for dealing with batch are close to what is often used in Active Learning. Theoretical results are interesting and useful, and the experiments are convincing for the dataset and method proposed. The paper is quite clear, especially for the reader familiar with the topic. Main concern: * The way I see it, the challenge with non-myopic setting is to choose next examples which would benefit the probabilistic model most (active learning) and at the same time will produce positive examples (search). This paper explores a certain approach to do that, but it is not clear it outperforms more naive and straightforward approaches, e.g. first performing Active Learning, and then search. This might be especially useful in the batch setting: by performing some sort of mix of examples in a batch: part of them go to AL, part to search. Having baselines like that (and method speed trade-offs) in experimental part would be more convincing, as well as having another dataset of a different nature. * Having more specific guidance on how to use theoretical guarantees in practice would be beneficial. * Is (5) submodular? is there proof? -- I have considered authors' comments, and they convinced me more in my score, as the authors performed more convincing experimentation and promised to add results to the paper.

Reviewer 2



This work investigates the different batch-mode extensions of an active search method called efficient nonmyopic policy (ENS) [12]. ENS achieves good performance efficiently because it assumes the sample selections are independent after a step [12]. This paper proposes two strategies: 1) converting the batch active search problem to sequential one by guessing the hidden labels of selected samples 2) try to enumerate all possible hidden labels of selected samples by Monte Carlo. Strength: Allowing to sample batches is important in practice. The work addresses several theoretical and practical challenges in many aspects of batch active search such as how difficult batch active search could be, why pessimistic oracle works well, and how to make the methods more efficient by pruning. The explanation is clear, and the literature review is comprehensive. The experiment results seem to show significant improvement as well. Weakness: The ideas of extension seem to be intuitive and not very novel (the authors seem to honestly admit this in the related work section when comparing this work with [3,8,9]). This seems to make the work a little bit incremental. In the experiments, Monte Carlo (batch-ENS) works pretty well consistently, but the authors do not provide intuitions or theoretical guarantees to explain the reasons. Questions: 1. In [12], they also show the results of GpiDAPH3 fingerprint. Why not also run the experiment here? 2. You said only 10 out of 120 datasets are considered as in [7,12]. Why not compare batch and greedy in other 110 datasets? 3. If you change the budget (T) in the drug dataset, does the performance decay curve still fits the conclusion of Theorem 1 well (like Figure 1(a))? 4. In the material science dataset, the pessimistic oracle seems not to work well. Why do your explanations in Section 5.2 not hold in the dataset? Suggestions: Instead of just saying that the drug dataset fits Theorem 1 well, it will be better to characterize the properties of datasets to which you can apply Theorem 1 and your analysis shows that this drug dataset satisfies the properties, which naturally implies Theorem 1 hold and demonstrate the practical value of Theorem 1. Minor suggestions: 1. Equation (1): Using X to denote the candidate of the next batch is confusing because it is usually used to represent the set of all training examples 2. In the drug dataset experiment, I cannot find how large the budget T is set 3. In section 5.2, the comparison of myopic vs. nonmyopic is not necessary. The comparison in drug dataset has been done at [12]. In supplmentary material 4. Table 1 and 2: why not also show results when batch size is 1 as you did in the drug dataset? 5. In the material science dataset experiment, I cannot find how large the budget T is set After rebuttal: Thanks for the explanation. It is nice to see the theorem roughly holds for the batch size part when different budgets are used. However, based on this new figure, the performance does not improve with the rate 1/log(T) as T increases. I suggest authors to replace Figure 1a with the figure in the rebuttal and address the possible reasons (or leave it as future work) of why the rate 1/log(T) is not applied here. There are no major issues found by other reviewers, so I changed my rate from tending to accept to accepting.

Reviewer 3



Summary: This work extends a line of work on active search to the batch setting. Some theory for this problem and its comparison to the sequential setting is derived and explored in addition to several relatively principled heuristics for the problem. Experiments on drug discovery datasets show gains over a natural baseline. Quality: The paper appears to be correct and the experiments are relatively comprehensive, at least for the narrow line of related work. Clarity: This paper is well-written and has a natural progression of ideas. Theorem 1 perhaps can be described intuitively better and Proposition 1 can be described a bit more precisely. Originality: This work is similar to an existing line of work in terms of experiments and results, but extends it to the batch setting. Significance: As the authors mention, trading off the inconvenience of sequential search and the statistical inefficiency of the batch setting is an important question which the authors address well. Additionally, this work extends effective heuristics from the sequential setting to the batch setting.